# Effectiveness of Mix-and-Match Vaccination in Preventing SARS-CoV-2 Omicron Variant Infection in Taiwan: A Test-Negative Control Study

**DOI:** 10.3390/vaccines11091441

**Published:** 2023-08-31

**Authors:** Yu-Tung Huang, Yi-Ching Chen, Chih-Hsien Chuang, Shang-Hung Chang, Cheng-Hsun Chiu

**Affiliations:** 1Center for Big Data Analytics and Statistics, Chang Gung Memorial Hospital, Taoyuan 333011, Taiwan; anton@cgmh.org.tw; 2Department of Health Care Management, Chang Gung University College of Management, Taoyuan 333323, Taiwan; 3Division of Pediatric Infectious Diseases, Department of Pediatrics, Chang Gung Memorial Hospital, Taoyuan 333423, Taiwan; cemily02@cgmh.org.tw; 4Molecular Infectious Disease Research Center, Chang Gung Memorial Hospital, Taoyuan 333423, Taiwan; doc1860@mail.sph.org.tw; 5Department of Epidemiology, Harvard T.H. Chan School of Public Health, Boston, MA 02215, USA; 6Department of Pediatrics, St. Paul’s Hospital, Taoyuan 330049, Taiwan; 7School of Medicine, College of Medicine, Fu-Jen Catholic University, New Taipei 242062, Taiwan; 8Division of Cardiology, Department of Internal Medicine, Chang Gung Memorial Hospital, Taoyuan 333423, Taiwan; afen@cgmh.org.tw; 9School of Medicine, Chang Gung University College of Medicine, Taoyuan 333323, Taiwan

**Keywords:** SARS-CoV-2 Omicron variant, mix-and-match vaccination, primary series, booster, vaccine effectiveness

## Abstract

This study aimed to evaluate the effectiveness (VE) of mix-and-match vaccination against SARS-CoV-2 Omicron variant infection and severe outcomes. An SARS-CoV-2 PCR-confirmed retrospective cohort from Chang Gung Medical System in Taiwan was constructed. Vaccination records were tracked from the National Immunization Information System and categorized by different regimens or unvaccinated status. The main outcomes are VE against PCR-confirmed infection and COVID-19-associated moderate to severe disease. Participants were observed during the Omicron wave from March to August 2022. Of 298,737 PCR testing results available, 162,219 were eligible for analysis. VE against infection was modest, ranging from 38.3% to 49.0%, while mRNA-based vaccine regimens revealed better protection against moderate to severe disease, ranging from 80.8% to 90.3%. Subgroup analysis revealed lower VE among persons with major illness in preventing moderate to severe disease. For young adults, the VE of protein-based vaccine regimens showed a comparable protection with other mixed vaccine regimens. The mix-and-match vaccination strategy provided modest clinical effectiveness in preventing Omicron variant infection. mRNA vaccine-based regimens were superior to other regimens against moderate to severe disease especially in older adults. The mix-and-match vaccination strategy could be an alternative to prevent COVID-19 in unstable vaccine supply regions.

## 1. Introduction

Different from any previous pandemic, the rapid development of COVID-19 vaccines modified the public health response to the pandemic and may have trimmed the path of viral evolution [1]. COVID-19 vaccines have been proven to be an efficient and cost-effective strategy in preventing disease-associated critical illness and death, in that at least a 63% reduction in global COVID-19-associated death was averted by vaccination during the first year of vaccine rollout [2]. However, following the rapid and wide use of vaccines, several issues were raised and remained important, needing further consideration even in the post-pandemic era, including rare but potentially life-threatening adverse events, difficulties in establishing the accurate correlates of protection (CoP) against current circulating lineages, and vaccine inequities.

Vaccine safety remains to be one of the most important issues. Some rare but potentially life-threatening adverse events associated with COVID-19 vaccination in certain age or gender populations were identified, such as thrombotic thrombocytopenic syndrome and Guillain–Barré syndrome following adenoviral vector-based vaccination and myocarditis/pericarditis following mRNA vaccination in adolescents and young adults [3,4,5,6,7]. Furthermore, although neutralizing antibodies are the current consensus in the indirect measurement of vaccine efficacy, an applicable CoP for the COVID-19 vaccine against the circulating variant of concern (VOC) remains still yet to be determined [8,9]. Real-world studies of vaccine effectiveness (VE) are therefore important from the public health perspective. Finally, a critical issue is the imbalance of vaccine distribution. Due to limited access to novel vaccine platforms, middle- to low-income countries had less chance to develop their vaccines, and many COVID-19 vaccines need special storage conditions and cold chain transportation, making the vaccine rollout in these countries even harder [10,11]. Therefore, a flexible strategy, such as the mix-and-match vaccination (mmV) strategy and easily accessible vaccines, was truly important to solve these problems. 

In Taiwan, there were limited vaccine resources in the early vaccine rollout stage. In early 2021, only ChAdOx1 nCOV-19 was available in Taiwan (since 21 March 2021), followed by mRNA-1273 which arrived three months later (since 8 June 2021), and BNT162b2 which was not available until September 2021 (since 22 September 2021). In addition to the adenoviral vector-based vaccines and mRNA vaccines, a protein-based vaccine, MVC-COV1901, was also granted emergency use authorization (EUA) by the Taiwan FDA (since 23 August 2021). MVC-COV1901 was a CpG1018 and aluminum hydroxide-adjuvanted SARS-CoV-2 pre-fusion-stabilized spike protein S-2P vaccine (Medigen Vaccine Biologics Corporation, Hsinchu, Taiwan) [12,13]. Given the limited vaccine resources and to maximize the vaccine distribution, the mmV strategy was suggested by the Advisory Committee on Immunization Practices (ACIP) of Taiwan. 

Previous studies have demonstrated the potential benefits of the mmV strategy, including better immunogenicity elicited by some regimens [14,15,16]. However, real-world effectiveness studies to assess the protection offered by different mixed regimens are scarce [17]. We therefore performed a retrospective observational study with a test-negative control design to evaluate the VE of protein-based vaccine-containing regimens and the mmV strategy in preventing COVID-19.

## 2. Methods

### 2.1. Data Sources

The data used in this study were retrieved from the Chang Gung Research Database (CGRD) and the National Immunization Information System (NIIS) dataset. CGRD is the largest electronic medical record (EMR)-based database, which includes seven branches with over 9000 beds, covering 6.1% of the outpatient service and 10.2% of the hospitalized patient service in Taiwan [18]. NIIS is an official dataset managed and maintained by the Taiwan Centers for Disease Control (CDC) [19]. All the COVID-19 vaccines were officially provided by Taiwan CDC, and all the utilization was entry-recorded into the NIIS in Taiwan. For persons receiving a vaccination overseas, it was an obligation for them to report their vaccination record to NIIS if they made a formal entry into Taiwan. 

We first collected the SARS-CoV-2 PCR testing records and screeners’ EMR data such as age, sex, location, major illness, admission, and death records from CGRD. Then, we linked the data with NIIS to obtain their vaccination records, including the vaccination date, vaccine brand, and order of vaccination. Both CGRD and NIIS datasets were encrypted and deidentified due to privacy concerns but linkable for research purposes. All data analyses were allowed on-site at the Taiwan CDC’s regulated workstation only.

### 2.2. Study Design and Population

Before 2022, Taiwan had an extremely low incidence of SARS-CoV-2 infection (6.25 cases per 10,000-person days), except for experiencing a small alpha wave in 2021. Not until mid-March 2023 was the Omicron variant imported into Taiwan’s local community, followed by wide transmission. Therefore, a retrospective SARS-CoV-2 PCR confirmation cohort was constructed to estimate mmV effectiveness against the SARS-CoV-2 infection between 1 March and 31 August 2022, when Taiwan was experiencing an endemic wave during the Omicron-variant-dominant period.

A conceptual schematic of our study design is presented in Appendix A. Those who underwent SARS-CoV-2 reverse transcription polymerase chain reaction (RT-PCR) testing during the above period at Chang Gung Medical Foundation (CGMF) hospital system were included in the study. There were two medical centers and four regional hospitals in the system that provided a SARS-CoV-2 PCR testing service during the study period; these hospitals also received PCR samples from test centers in the local community. We retrospectively reviewed their vaccination records and post-vaccination status to explore mmV effectiveness. For cases with multiple testing circumstances, we only included the first record for positive cases and the last record for negative cases. We identified the screened date as the observed end date, and 1 March 2022 was the start date of observation. Because some COVID-19 vaccines were approved only for use in persons 20 years of age or older in Taiwan, persons under this age were excluded. Moreover, persons without complete vaccination, defined as receiving less than three doses of COVID-19 vaccine before 1 March and those with missing data, were also excluded.

We included unvaccinated persons to form an unvaccinated cohort to estimate mmV effectiveness. Some persons received PCR testing for non-medical needs (e.g., border immigration or emigration), and most of them had only a short-term stay in Taiwan. Therefore, in the unvaccinated group, we included those who had at least a one-time visit to the Chang Gung Medical Foundation hospital system and no vaccination records.

### 2.3. Definition of Mix-and-Match Vaccination

There were five vaccines, including ChAdOx1 nCOV-19, mRNA-1273, BNT162b2, MVC-COV1901, and NVX-CoV2373 recommended by the Advisory Committee on Immunization Practices (ACIP) of the Ministry of Health and Welfare after being granted for emergency use authorization by the Taiwan Food and Drug Administration (FDA). However, a few people have been vaccinated abroad with Taiwan FDA-approved and/or nonapproved vaccines such as vaccines from Johnson & Johnson, Sinopharm, Sinovac, Bharat, and so on. A total of 49 types of mmV were found in our data. Considering the sample size of each type, vaccine variety, and the study aims, we classified mmV into ten major types. These types included non-mmV (MMM: three doses of mRNA-1273, BBB: three doses of BNT162b2, MVC-3-dose: three doses of MVC-COV1901), same-platform mmV (BBM: two doses of BNT162b2 followed by mRNA-1273 as booster, MMB: two doses of mRNA-1273 followed by BNT162b2 as booster), and cross-platform mmV (AAM: two doses of ChAdOx1 nCOV-19 followed by mRNA-1273 as booster, AAB: two doses of ChAdOx1 nCOV-19 followed by BNT162b2 as booster, AMM: heterologous primary series with one dose of ChAdOx1 nCOV-19 and one dose of mRNA-1273 followed by mRNA-1273 as booster, MVC primary series: two doses of MVC-COV1901 followed by any other COVID-19 vaccine as booster, MVC booster: non-MVC-COV1901 vaccine as primary series followed by MVC-COV1901 as booster). All other mmV types were excluded due to their small sample size and miscellany.

### 2.4. Definition of Outcome and Other Variables

The primary outcome of the study is SARS-CoV-2 infection confirmed by positive RT-PCR testing, irrespective of being symptomatic or not. Persons with confirmed SARS-CoV-2 infection within seven days after receiving vaccination were excluded. The secondary outcome is the severity of the illness at a moderate or severe level (M/S cases), which was defined by either the use of remdesivir or hospitalization due to COVID-19, according to the definition by the Taiwan CDC. During the study period, COVID-19 patients could only be hospitalized if they needed specific medical care because of the limited health care capacity available. Most of the mildly infected cases self-isolated at home or were sent into collective quarantine provided by a local health authority. The outcome period of the study was 1 March to 31 August 2022.

Other vaccinee characteristics such as age, sex, region, health status, and time to vaccination were also included in this study. Age was divided into four groups: 20–49, 50–64, 65–74, and above 75 years. Because the COVID-19 pandemic was different by geographic region, the “South Region” was defined when cases were enrolled from southern Taiwan, including Kaohsiung, Yunlin, and Chiayi; and the “North Region” was for cases enrolled from northern Taiwan, including Kee-Lung, Taipei, and Taoyuan. For persons with underlying health conditions, we employed the status with/without major illness, such as cancer, chronic renal failure, autoimmune disease, congenital major disease, major trauma or burn, organ transplant, long-term dependence disease, et al., as proxy indicators which defined persons who had major illness according to the National Health Insurance Administration of Taiwan. In addition, considering that the vaccine protection would wane overtime and each person would have a different duration from the third vaccination to the beginning of our observation (shown as the orange line in Appendix A), we set up the duration time (week) variable from the third vaccination date to the start date to adjust the entry status. To simplify this variable, we categorized it into three groups: shorter than four weeks, four to eight weeks, and longer than eight weeks.

### 2.5. Statistical Analyses

The eligible persons were described with the frequency, with percentage and mean with standard deviation. For mmV group comparison, the crude incidence rate of breakthrough infection per 1000 person-days, admission rate due to breakthrough infection per 100,000 person-days, and admission proportion for breakthrough infection cases were calculated. Moreover, the adjusted hazard ratios (HR) comparing the incidence of breakthrough infection in each mmV group were obtained via a Cox proportional hazard model. The VE and 95% confidence intervals (CI) for each group were estimated as one minus the adjusted HR and are presented as a percentage.

### 2.6. Research Ethics and Oversight

This study was conducted in accordance with the Declaration of Helsinki and the Declaration of Taipei on ethical considerations regarding health databases by the World Medical Association. The study was approved by the Institutional Review Board of Chang Gung Medical Foundation (IRB No: 202201645B0) and the Taiwan CDC. Because all data converted from the CGRD and the NIIS were encrypted, deidentified, and under tight regulations for on-site analysis by the Taiwan CDC, informed consent was waived. All analysis work was discussed at the workstation of the Taiwan CDC. Statistical analysis was performed using the SAS software version 9.4 (SAS Institute, Cary, NC, USA). A two-sided *p*-value < 0.05 was considered statistically significant.

## 3. Results

### 3.1. Baseline Demographics

Between 1 March 2022 and 31 August 2022, a total of 298,737 PCR testing results were available in CGRD. After excluding unmatched vaccination records and the vaccinees under 20 years old, there were 266,784 eligible persons, including 236,098 receiving at least one dose of a vaccination and 30,686 unvaccinated persons (Figure 1). Among the 236,098 vaccinated persons, we further excluded persons with incomplete vaccination, the interval between PCR testing and the third dose vaccination being less than seven days, missing/incomplete data, and vaccination regimens that were not our targets for comparison in this study.

Regarding baseline age distribution, the mean age among those with different vaccinations ranged from 40 to 60 years old, except for the older mean age in the MMM group [Mean (Standard deviation): 63.44 (13.83)] (Appendix A). The interval between individuals who received the third dose of the COVID-19 vaccine to the start date of observation (1 March 2022) is similar, ranging from 4.07 (2.08) to 5.61 (2.37) weeks, except in the BBM and BBB groups [1.73 (1.14) to 1.95 (1.88) weeks] (Appendix A). This is because of the late rollout of the BNT vaccine in Taiwan.

In the final analysis, 28,538 PCR-confirmed infections were detected during the study period, with a total of 133,681 test-negative controls. The baseline demographics between the test-negative and test-positive groups are shown in Table 1. Variations between the groups were found, such as age distribution, region, and vaccination intervals. Persons with major illness were less infected than those without. Furthermore, compared to the vaccinated cohort, persons in the unvaccinated cohort showed a higher proportion with major illness (Appendix A). 

### 3.2. Vaccine Effectiveness against Infection

In this study, the incidence of SARS-CoV-2 infection ranged from 1.5 to 2.2 per 1000 persons per day in different vaccination regimens and was 2.9 per 1000 persons per day in the unvaccinated group (Table 2). Compared to the unvaccinated group, VEs against SARS-CoV-2 infection were similar, and ranged from 38.3% to 49.0% (Figure 2). Regimens with three doses of mRNA vaccine had slightly higher effectiveness (44.5–49.0%), followed by the regimen of primary series with a viral vector vaccine and an mRNA vaccine booster (42.4–42.7%). The protein-based vaccine as either the primary series or booster also offered similar protection among the groups (38.3–44.8%). The estimated vaccine protection in the AMM group, although the regimen contained two doses of mRNA vaccine, was relatively lower than for the other mRNA vaccine groups (39.9%).

### 3.3. Vaccine Effectiveness against COVID-19-Associated Moderate to Severe Disease

The crude incidence of COVID-19-associated moderate to severe disease following SARS-CoV-2 infection during the study period is highest in the unvaccinated group, with 48.2 per 100,000 persons per day, and it is much lower in the vaccinated group, ranging from 0.91 to 10.37 per 100,000 persons per day (Table 2). Compared to the protection against infection, VE against moderate to severe disease is much more robust (Figure 2; Appendix A). Effectiveness against COVID-19-associated moderate to severe disease could reach nearly 70% in each group, except for those receiving the protein-based vaccine as a booster (66.1%). Three doses of mRNA vaccination still provided optimal protection among groups (80.8–90.3%), followed by the protection by the viral vector vaccine in the primary series with an mRNA vaccine boosting, and protein-based vaccine regimens (69.7–79.6%). The estimated VE revealed a high protection in the AMM group (91.4%), but it may be due to a significantly younger mean age [40.74 (12.21)] of the vaccinees in this group (Appendix A).

### 3.4. Subgroup Analysis according to Underlying Major Illness

Subgroup analysis according to underlying major illness was performed as a sensitivity test. In terms of protection against infection, an analysis of data from persons without any major illness revealed consistent results with those obtained from the overall pooled analysis (Figure 2 and Figure 3; Appendix A). For persons with major illness, the results were more heterogenous with generally widened confidence intervals because of the relatively small sample sizes (Figure 3; Appendix A). VE was similar to that obtained from the pooled analysis, except in those who received three doses of BNT162b2 and heterologous primary series with an mRNA-1273 booster; these groups showed relatively lower vaccine protection compared to other groups.

Compared to the pooled analysis, vaccine protection against moderate to severe disease (63.1–92.5%) was similar in persons without major illness, and the vaccine regimen with a protein-based vaccine as a booster yielded the lowest VE (63.1%) (Figure 3; Appendix A). On the other hand, although results were heterogenous due to limited sample sizes, the effectiveness against moderate to severe disease was generally lower in persons with major illness in the sub-group analysis (Figure 2 and Figure 3; Appendix A). This finding is consistent with results from other real-world effectiveness studies as well as immunogenicity studies that populations with underlying major illness may have attenuated immune responses to vaccination, reflecting less vaccine protection in the real-world setting.

### 3.5. Subgroup Analysis according to Different Age Groups

Regarding age being one of the significant variables contributing to the different risks of COVID-19-associated severe disease, we further analyzed VE in different age groups. For preventing SARS-CoV-2 Omicron variant infection, three doses of mRNA vaccines still provide more robust protection in older adults aged more than 75 years old (48.0–88.2%) (Appendix A). On the other hand, the estimated effectiveness against infection was similar among other vaccination regimens in different age groups (Appendix A). However, because the number of persons receiving the protein-based vaccine was relatively small in the older adults, our study could not assess protection against infection properly. For the young adult population with age ranging from 20 to 49 years old, the effectiveness against infection of protein-based vaccine regimens was comparable among groups.

For persons who received mRNA-1273 as the primary series, despite being significantly older [Mean age of MMM group: 63.4 (13.8) years old; MMB group: 58.0 (15.8) years old] than those with other vaccination regimens, the vaccination still offered the most substantial protection against moderate to severe disease in older adults. However, for persons who received two doses of viral vector vaccine followed by an mRNA booster, although the effectiveness in those below 65 years old was satisfactory (77.3–86.2%), it decreased significantly in the population over 65 years of age (Appendix A). The effectiveness against COVID-19-associated moderate to severe disease of protein-based vaccines was comparable to other vaccination regimens among young adults between 20 and 49 years of age.

## 4. Discussion

Compared to the studies that only focused on one or two vaccine platforms [20,21,22], we analyzed the effectiveness of different vaccination regimens, including a protein-based vaccine. Although the immunogenicity studies of the mmV strategy revealed superiority in certain regimens [14,15,16], our results showed similar protection in preventing documented SARS-CoV-2 Omicron variant infection among different regimens. The estimated effectiveness against SARS-CoV-2 variant infection was compatible with data from other studies, which, however, is generally lower than the effectiveness against Delta variant infection [23,24,25,26]; such decreased effectiveness reflected the SARS-CoV-2 Omicron variant’s dominant ability of immune evasion. The discordant results of immunogenicity and real-world effectiveness indicated that establishing the CoP focusing on current circulating variants would be important but difficult.

While the effectiveness against infection yielded no significant difference among different regimens, mRNA-based vaccination revealed better protection against COVID-19-associated moderate to severe disease than viral vector and protein-based vaccines. In this study, the effectiveness achieved 80.8–90.3% in those who received three doses of mRNA vaccines. The result is in accordance with previous studies using mRNA vaccines and evaluating the same endpoint [27,28]. For persons with underlying major illness, VE was lower; such an observation, in fact, can be extrapolated from the immunogenicity data that the populations with underlying major illness may have attenuated immune responses to vaccination, leading to less protection in the real-world setting [29,30].

Due to limited distribution, the effectiveness of protein-based vaccination regimens was less reported. This is one of the leading studies examining the real-world effectiveness of a protein-based vaccine, MVC-COV1901. In terms of the effectiveness against documented SARS-CoV-2 Omicron variant infection, the protection offered by MVC-COV1901 was similar to the mRNA or viral vector vaccines. Although the effectiveness against moderate to severe disease is relatively inferior to mRNA-vaccine-based regimens, it still provides good protection. Considering such comparable effectiveness with other vaccination regimens and a relatively higher incidence of mRNA-vaccine-associated local or systemic reactogenicity, a protein-based vaccine may still be one of the alternative choices for specific populations, for example, persons with contraindication to receive an mRNA vaccine or persons at high risk for mRNA-vaccine-associated severe adverse events.

Our study has several limitations. The major limitation is the uneven distribution of case numbers among groups. The sample sizes in some vaccine groups were relatively small, which may cause a widened confidence interval regarding the effectiveness. This also limited the possibility to access vaccine protection over time in groups with a small sample size. This condition is inevitable because of the vaccine rollout plan in Taiwan. During the early stage, Taiwan residents received COVID-19 vaccines based on a sequential priority according to their risk for severe COVID-19. When the booster dose was suggested, vaccine resources were more sufficient, so the government recommended a flexible mmV strategy for the booster program, leading to a large variety of vaccination regimens being used in Taiwan. Although age distribution was uneven among the study groups, we performed a subgroup analysis to correct the bias. The subgroup analysis showed that mRNA-based vaccination regimens provided better protection against moderate to severe disease than protein-based or viral vector vaccines in older adults, which is consistent with results obtained from the overall pooled data analysis. However, no difference was observed in the young adult population.

## 5. Conclusions

During the SARS-CoV-2 Omicron variant wave in Taiwan, the mmV strategy provided modest effectiveness in preventing Omicron variant infection in the real-world setting. mRNA-based vaccination regimens were still preferred against moderate to severe disease, especially in older adults, while for those at high risk for mRNA-vaccine-associated severe adverse events, a protein-based vaccine that showed comparable protection could be an alternative. 

## Figures and Tables

**Figure 1 vaccines-11-01441-f001:**
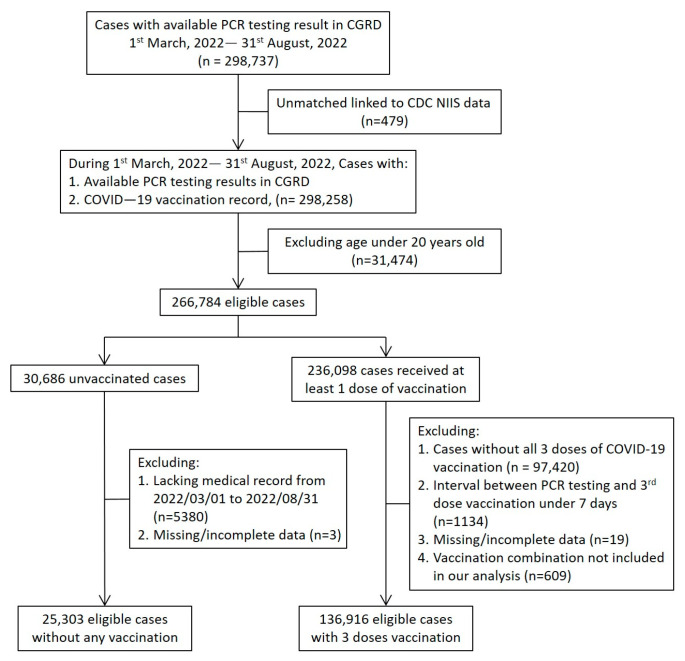
Study population and algorithm. Note: The data used in the study were retrieved from the Chang Gung Research Database (CGRD) and the National Immunization Information System (NIIS) dataset between 1 March and 31 August 2022 in Taiwan.

**Figure 2 vaccines-11-01441-f002:**
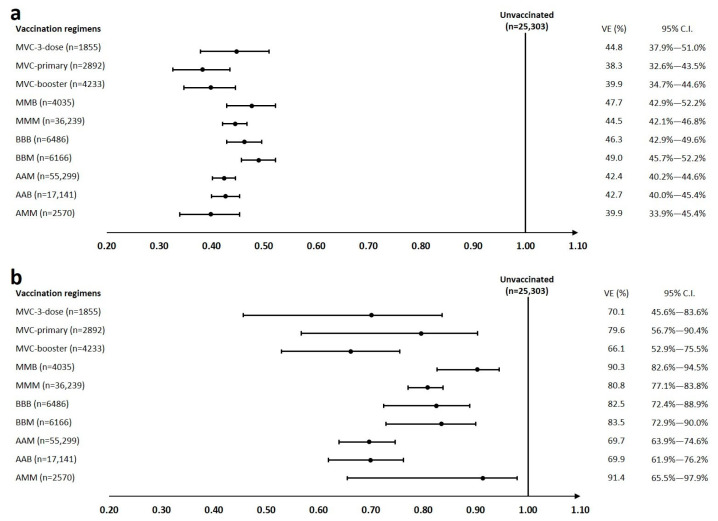
Forest plot of vaccine effectiveness against SARS-CoV-2 Omicron variant infection and COVID-19-associated moderate to severe disease. (**a**,**b**) show the effectiveness against SARS-CoV-2 Omicron variant infection and COVID-19-associated moderate to severe disease, respectively. Note: The vaccine effectiveness and 95% confidence intervals for each group were estimated as one minus the adjusted hazard ratio and are presented as a percentage. VE stands for vaccine effectiveness and C.I. for confidence interval. Abbreviations: A, ChAdOx1 nCOV-19; M, mRNA-1273; B, BNT162b2; MVC, MVC-COV1901.

**Figure 3 vaccines-11-01441-f003:**
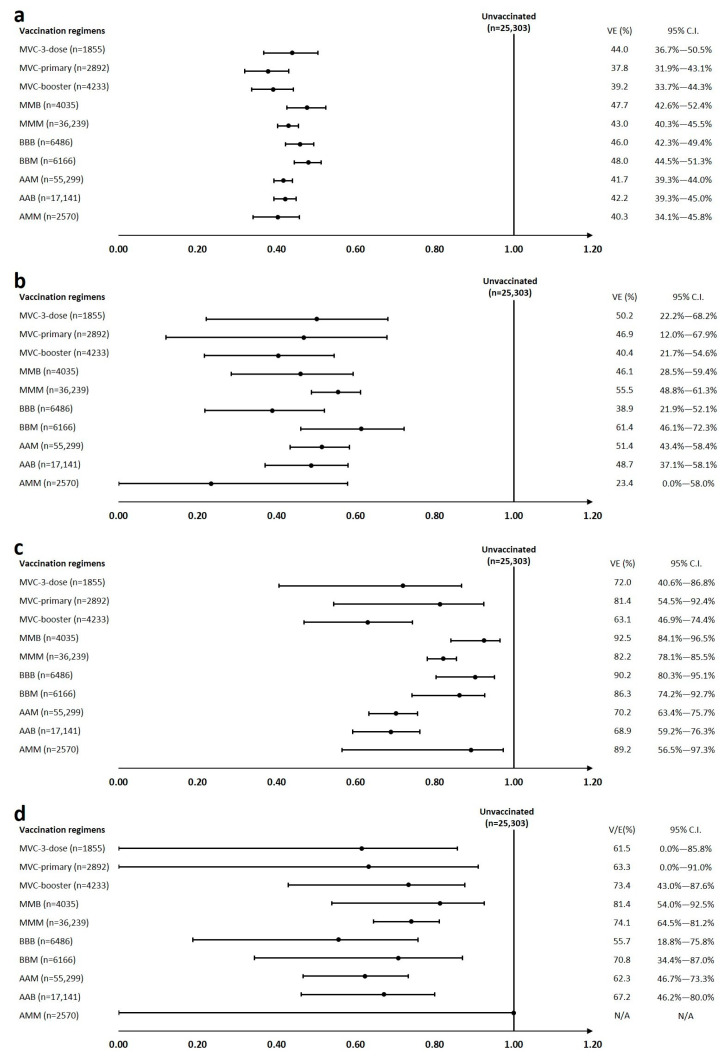
Subgroup analysis of the vaccine effectiveness against Omicron variant infection and COVID-19-associated moderate to severe disease. (**a**) shows effectiveness against Omicron variant infection in persons without underlying major illness, (**b**) shows effectiveness against Omicron variant infection in persons with major illness, (**c**) shows effectiveness against COVID-19-associated moderate to severe disease in persons without major illness, and (**d**) shows effectiveness against COVID-19-associated moderate to severe disease in persons with major illness. Note: The vaccine effectiveness and 95% confidence intervals for each group were estimated as one minus the adjusted hazard ratio and are presented as a percentage. VE stands for vaccine effectiveness and C.I. for confidence interval. Abbreviations: A, ChAdOx1 nCOV-19; M, mRNA-1273; B, BNT162b2; MVC, MVC-COV1901.

**Table 1 vaccines-11-01441-t001:** Characteristics of study populations.

Characteristic	Total (n = 162,219)		Test Negative (n = 133,681)		Test Positive (n = 28,538)		ASMD
	n	%	n	%	n	%	
Survival							0.27
Survived	161,864	99.78	133,681	100.00	28,183	98.76	
Died	355	0.22	0	0	355	1.24	
Age group							0.24
20–49	67,490	41.60	52,095	38.97	15,395	53.95	
50–64	45,934	28.32	39,431	29.50	6503	22.79	
65–74	30,642	18.89	26,848	20.08	3794	13.29	
75+	18,153	11.19	15,307	11.45	2846	9.97	
Sex							0.01
Female	92,341	56.92	76,267	57.05	16,074	56.32	
Male	69,878	43.08	57,414	42.95	12,464	43.68	
Region							0.34
South ^a^	66,861	41.22	58,998	44.13	7863	27.55	
North ^b^	95,358	58.78	74,683	55.87	20,675	72.45	
Major illness ^c^							0.14
without	141,067	86.96	115,123	86.12	25,944	90.91	
with	21,152	13.04	18,558	13.88	2594	9.09	
Interval time ^d^							0.18
<4 weeks	74,380	45.85	58,929	44.08	15,451	54.14	
4–8 weeks	82,307	50.74	70,164	52.49	12,143	42.55	
>8 weeks	5532	3.41	4588	3.43	944	3.31	

Abbreviation: ASMD, absolute standardized mean difference. ^a^ Cases enrolled from southern Taiwan (Kaohsiung, Yunlin, and Chiayi). ^b^ Cases enrolled from northern Taiwan (Kee-lung, Taipei, New Taipei City, Taoyuan). ^c^ Persons who hold major systemic diseases issue according to the National Health Insurance Administration. ^d^ The duration between the days when vaccinees received the 3rd dose of COVID-19 vaccine and 1 March 2022.

**Table 2 vaccines-11-01441-t002:** Crude incidence rate of primary and secondary outcomes among groups.

Vaccine Group	Total Vaccinees	Total Follow-Up Time ^a^	Test Positive Cases	M/S Cases	Crude IR for Infection ^b^	Crude IR for M/S Disease ^c^	M/S Disease (%)
Unvaccinated	25,303	2,344,473	6817	1130	2.91	48.20	16.58
MVC-3-dose	1855	169,615	289	11	1.70	6.49	3.81
MVC-primary	2892	258,425	580	7	2.24	2.71	1.21
MVC-booster	4233	385,590	639	40	1.66	10.37	6.26
MMB	2570	219,004	466	2	2.13	0.91	0.43
MMM	36,239	3,358,518	4882	266	1.45	7.92	5.45
BBB	6486	584,824	1157	19	1.98	3.25	1.64
BBM	6166	553,906	1108	16	2.00	2.89	1.44
AAM	55,299	4,939,052	9099	362	1.84	7.33	3.98
AAB	17,141	1,521,900	2948	100	1.94	6.57	3.39
AMM	4035	369,188	553	12	1.50	3.25	2.17

Abbreviation: IR, incidence rate; M/S, moderate or severe. ^a^ Person-days. ^b^ Per 1000 person-days. ^c^ Per 100,000 person-days.

## Data Availability

The data presented in this study are available on request from the corresponding authors.

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
