# Peer review of "Effectiveness of Mix-and-Match Vaccination in Preventing SARS-CoV-2 Omicron Variant Infection in Taiwan: A Test-Negative Control Study"

_vaccines, 2023, doi:10.3390/vaccines11091441_

Round 1

Reviewer 1 Report

1. In this sty, the authors described the effectiveness of mix and match COVID-19 vaccines against Omicron variants of SRAS-CoV-2 Virus. One important question is all the RT-PCR positive cases were omicron variants? How did the authors confirmed that all cases were omicron variants? Please describe confirmation tests that showed all were Omicron variants at the revised manuscript.

2. Regarding the study design, the authors described as retrospective cohort study in abstract but the authors also described again with retrospective case control at Page -2 , line no-80. Which one is the most suitable one for your study design? How many control cases were enrolled? To describe case-control stud. All the control were comparable with the cases?

3. The authors categorized different kinds of mix and max vaccination strategies. We also noted some homologus booster vaccination in this study such as MMM, BBB, etc. For clear understanding please also categorized which groups were homologus and which were heterologous booster vaccination in this study at the revised manuscript.

4. The reviewer noted the authors excluded 609 people due to the reasons of vaccination combination not included at your study. What are those combinations and why did the authors did not add and describe the brief description at your revised manuscript.

5.Regarding the severity of the disease, please add the operational definition for diagnosing moderate or severeCOVID-19 disease in this study.

6.At the conclusion, the authors described that protein based vaccine is less reactogenicity to other types. In this study, the authors did not find out the safety and immunogenicity of the protein-based vaccines and the authors did not ask the participants for choosing the types of vaccines. It is better to remove those words at your conclusion.

7. The authors described that this study was the first study about the effectiveness of the protein based vaccines. Is it really  first study?

Author Response

Please check the attached files for our reply to your comments and the revised manuscript based on all reviewers' comments.

Reviewer 2 Report

The authors attempted to evaluate the effectiveness various vaccinations against SARS-CoV-2 Omicron infection and severe outcomes. Given the limited vaccine resources and to maximize the vaccine distribution, the mix-and-match vaccination strategy was performed in Taiwan. Therefore, it was possible for them to do real-world effectiveness studies to assess the protection offered by various regimens including a protein-based vaccine. In this respect, this study is very unique. Although mRNA vaccine-based regimens were superior to other regimens against moderate to severe disease in the older adults, mix-and match vaccination strategy provide modest clinical effectiveness. Interestingly, the current study shows that a protein-based vaccine provides a good protection although the effectiveness against moderate to severe disease is relatively inferior to mRNA vaccine-based regimens, suggesting a protein-based vaccine may still be one of the alternative good choices for specific populations. The manuscript is well-written, and the data is very useful. I have no serious criticisms except for Table 2. Table 2 does not seem relevant to the current study.

Author Response

(The authors gave the same response as above.)

Reviewer 3 Report

The authors have put together a manuscript detailing the vaccine efficacy of various mix and match vaccination regimes used in Taiwan during the pandemic. Importantly, the manuscript details the efficacy of a locally produced protein based vaccine (MVC-COV1901). Overall the manuscript reads well, despite several typos. However, there is a glaring mistake with Table 2, as the wrong Table 2 was included in the manuscript.

Listed are comments for the authors to address:

Line 102: Was Omicron only imported into Taiwan mid-March 2023? or is that a typo (and meant to be 2022)? The following line describes that Omicron was endemic between March and August 2022.

Line 228-230, 251-254: It appears that the wrong table was included for Table 2? The existing table 2 describes "Diversity profiles of Salmonella isolates based on MLST, serotyping, and antimicrobial resistance." and is not relevant to the incidence or severity of SARS-CoV-2 infection?

Line 235-237: Could this be attributed to the age groups involved as MVC-primary also showed lower estimated protection (38.3%) and was also made up of predominantly 20-40s (>70%).

Figure 2-3; Supp Figure 2-3: The confidence intervals for some of these cohorts are very wide, and that is partially attributed to the small sample sizes involved in some of these subgroup analysis. As such, it is useful to the reader if the (n) for each subgroup is listed beside the different vaccination regimens for these forest plots, instead of having to refer to Supplementary Table 1 to get that information.

Line 333: Unclear why "protein-based" is in bold.

The manuscript reads well, despite several typos. The authors should read through it once more to address these changes.

Author Response

(The authors gave the same response as above.)

Round 2

Reviewer 1 Report

The author responded well on comments. I have no more comments.